# Deep Learning-Based Surgical Treatment Recommendation and Nonsurgical Prognosis Status Classification for Scaphoid Fractures by Automated X-ray Image Recognition

**DOI:** 10.3390/biomedicines12061198

**Published:** 2024-05-28

**Authors:** Ja-Hwung Su, Yu-Cheng Tung, Yi-Wen Liao, Hung-Yu Wang, Bo-Hong Chen, Ching-Di Chang, Yu-Fan Cheng, Wan-Ching Chang, Chu-Yu Chin

**Affiliations:** 1Department of Computer Science and Information Engineering, National University of Kaohsiung, Kaohsiung 81148, Taiwan; bb0820@ms22.hinet.net; 2Department of Diagnostic Radiology, Kaohsiung Chang Gung Memorial Hospital, Chang Gung University College of Medicine, Kaohsiung 83301, Taiwan; haaman010@hotmail.com (C.-D.C.); prof.chengyufan@gmail.com (Y.-F.C.); o927003551@gmail.com (W.-C.C.); 3Department of Intelligent Commerce, National Kaohsiung University of Science and Technology, Kaohsiung 82444, Taiwan; pinkwen923@gmail.com (Y.-W.L.); fuzhu325@gmail.com (H.-Y.W.); 4Department of Information Management, National Kaohsiung University of Science and Technology, Kaohsiung 82445, Taiwan; f111118122@nkust.edu.tw; 5Telecommunication Laboratories Chunghwa Telecom Company Limited, Kaohsiung 80002, Taiwan; cchuyu@cht.com.tw

**Keywords:** biomedical image, scaphoid fracture recognition, surgical treatment recommendation, nonsurgical prognosis classification, deep learning

## Abstract

Biomedical information retrieval for diagnosis, treatment and prognosis has been studied for a long time. In particular, image recognition using deep learning has been shown to be very effective for cancers and diseases. In these fields, scaphoid fracture recognition is a hot topic because the appearance of scaphoid fractures is not easy to detect. Although there have been a number of recent studies on this topic, no studies focused their attention on surgical treatment recommendations and nonsurgical prognosis status classification. Indeed, a successful treatment recommendation will assist the doctor in selecting an effective treatment, and the prognosis status classification will help a radiologist recognize the image more efficiently. For these purposes, in this paper, we propose potential solutions through a comprehensive empirical study assessing the effectiveness of recent deep learning techniques on surgical treatment recommendation and nonsurgical prognosis status classification. In the proposed system, the scaphoid is firstly segmented from an unknown X-ray image. Next, for surgical treatment recommendation, the fractures are further filtered and recognized. According to the recognition result, the surgical treatment recommendation is generated. Finally, even without sufficient fracture information, the doctor can still make an effective decision to opt for surgery or not. Moreover, for nonsurgical patients, the current prognosis status of avascular necrosis, non-union and union can be classified. The related experimental results made using a real dataset reveal that the surgical treatment recommendation reached 80% and 86% in accuracy and AUC (Area Under the Curve), respectively, while the nonsurgical prognosis status classification reached 91% and 96%, respectively. Further, the methods using transfer learning and data augmentation can bring out obvious improvements, which, on average, reached 21.9%, 28.9% and 5.6%, 7.8% for surgical treatment recommendations and nonsurgical prognosis image classification, respectively. Based on the experimental results, the recommended methods in this paper are DenseNet169 and ResNet50 for surgical treatment recommendation and nonsurgical prognosis status classification, respectively. We believe that this paper can provide an important reference for future research on surgical treatment recommendation and nonsurgical prognosis classification for scaphoid fractures.

## 1. Introduction

### 1.1. Background

Recent advances in artificial intelligence have enabled a large improvement for biomedical sciences. In particular, for biomedical image recognition, deep learning is widely used as a solution for detecting cancers and diseases. In the field of biomedical image recognition, scaphoid fracture recognition is a hot topic because scaphoid fractures are not easy to detect. In practice, conventional radiographs are initially negative in 20–30% of scaphoid fractures on first presentation [1]. That is, a considerable number of such fractures are not immediately obvious after trauma and even radiologically invisible until several weeks after the original injury. Therefore, an early diagnosis can prevent a misjudgement and a delay in treatment. This is why automated scaphoid fracture recognition is so popular that a number of related studies were proposed on this topic. Even though recent scaphoid fracture recognition has achieved significant success, there remains a further issue of making a surgical treatment recommendation. In real applications, a nonsurgical treatment is an important decision for the prognosis because a nonsurgical treatment will lead to different future statuses such as avascular necrosis, non-union and union [2]. Therefore, how to effectively recognize a nonsurgical prognosis from images is another issue ignored by recent studies. To address these issues, in this paper, we provide solutions for surgical treatment recommendations and nonsurgical prognosis status classification by automated X-ray image recognition. First, a foundational operation named scaphoid segmentation is performed. Next, fracture recognition is executed to filter the positive images by fracture recognition. Then, for positive patients, a surgical treatment recommendation is achieved by further recognition of positive segmentations. Even if the fracture is not obvious, the doctor can still make an effective treatment decision to opt for surgery or not. Moreover, for a follow-up visit, if the patient does not require surgery, the current prognosis status is classified into one of the statuses of avascular necrosis, non-union and union. Here, the nonsurgical prognosis image classifier provides an effective support in classifying nonsurgical prognosis images. Although this study’s aims were a surgical treatment recommendation and nonsurgical prognosis status classification, the scaphoid segmentation and fracture recognition were very important in this study. Without a successful scaphoid segmentation and fracture recognition, a high-quality surgical treatment recommendation and nonsurgical prognosis status classification are not easy to achieve. Figure 1 shows the successfully segmented image examples of surgery, no surgery, avascular necrosis, non-union and union. Images (a) and (b) show different displaced states, while (c) and (d) indicate the worst unexpected cases. Image (e) depicts a recovered status of union. In practice, the references for a surgical treatment rely heavily on the appearance of a distal fracture and the waist scaphoid with no or slight displacements.

Overall, the contribution of this study can be summarized as follows.

I.For novelty, although many related studies have proposed effective methods for scaphoid fracture recognition, no method was proposed for surgical treatment recommendation and nonsurgical prognosis status classification for scaphoid fractures. However, the right treatment can effectively prevent unexpected results such as avascular necrosis and non-union. Hence, this paper recommends a solution for making an effective surgical decision by deep learning-based image recognition. Also, in this paper, the nonsurgical prognosis image classification is proposed to help the radiologist effectively recognize the nonsurgical prognosis status.II.For robustness, the best solutions for surgical treatment recommendation and nonsurgical prognosis image classification were elicited by comprehensive effectiveness assessments of recent convolutional neural networks, including ResNet [3], DenseNet [4], InceptionNet [5] and EfficientNet [6].III.For technique, based on recent convolutional neural networks, data augmentation and transfer learning were integrated to increase the prediction precision. Also, the network architecture was revised to improve the quality.IV.For practical application, the approximated models were tested on a real dataset provided by the Kaohsiung Branch of Chang Gung Hospital, Taiwan. These models will be applied to the other branches of Chang Gung Hospital in Taiwan.

### 1.2. Related Work

Because this paper includes several main modules with respect to scaphoid segmentation, fracture recognition, surgical treatment recommendation and nonsurgical prognosis status classification, the related works will be summarized into the aspects of object detection and fracture image recognition in this section. Further, a comparative study on recent scaphoid fracture recognition studies will also be performed.

#### 1.2.1. Object Detection Using Deep Learning

Over the past few decades, deep learning has been successfully applied to many multimedia fields such as multimedia recognition [7], multimedia recommendations [8], multimedia generation [9] and so on. In particular, there have been several applications of deep learning in the field of computer vision (CV), including image classification [10], object detection [11] and image segmentation [12]. Image segmentation classifies the pixels into an instance group, while object detection detects the objects in an image using bounding boxes. In this study, we adopted object detection YOLO (You Only Look Once) models instead of image segmentation to segment the scaphoid object (called scaphoid segmentation in this paper) in an X-ray image. YOLO is an anchor-based one-stage object detection method where the input is an image, and the output will be several bounding boxes and the desired labels. YOLO has been upgraded into eight versions. YOLO-v1 [13] partitions an image into a set of grids. For each grid, it predicts the confidences of two bounding boxes. For each box, it calculates the probability of each category. Finally, it uses Non-Maximum Suppression (NMS) to filter bounding boxes. YOLO-v1 can only detect 7 × 7 objects at most. Afterwards, YOLO-v2 [14] was extended to Darknet-19, and the grid number was increased to 13 × 13. According to *k* anchor boxes, a total of 13 × 13 × *k* objects can be detected. Further, the calculation cost decreased by 33% in contrast to YOLO-v1 and the top 5 accuracy increased from 88% to 91.2%. YOLO-v3 [15] was proposed with a new network Darknet-53 and replaced softmax with a logistic classifier. Further, it uses a feature pyramid network (FPN) to make predictions on three different scales and then filters features from these scales. YOLO-v4 [16] extends Darknet-53 into CSPDarknet-53 as the backbone. Also, it improved the fusion of local features and global features, thereby enriching the representations of the final feature maps. After YOLO-v4, versions 5 to 8 [17,18,19,20] were proposed to improve the effectiveness and efficiency of object detection.

#### 1.2.2. Fracture Image Recognition

Using object detection, bones can be segmented. Accordingly, a set of previous works were focused on fracture image recognition [21,22,23]. Because our aim is scaphoid fracture recognition, here, we only listed the recent studies on scaphoid fracture recognition. Tung et al. [24] conducted a detailed empirical approximation to pursue high-quality scaphoid fracture recognition in X-ray images. Based on CT (Computed Tomography) and MRI (Magnetic Resonance Imaging), Yoon et al. [25] employed Cascade R-CNNs (Region-Based Convolutional Neural Networks) to filter the scaphoid first. Then, EfficientNetB3 was used to recognize the fracture. Li et al. [26] achieved scaphoid segmentation and fracture recognition using YOLOv3 and MobileNetV3 [27], respectively. Singh et al. [28] proposed a residual-based convolutional neural network to achieve scaphoid fracture recognition. Hendrix et al. [29] adopted YOLOv5 and InceptionNetV3 to accomplish scaphoid segmentation and fracture recognition, respectively, using multi-view radiographs of the hand and wrist. Yang et al. [30] used Faster R-CNN [31] to segment the scaphoid. Then, a feature pyramid network and attention mechanism were embedded into ResNet152 for fracture recognition. Ozkaya et al. [32] compared the performances of ResNet50, an ED (Emergency Department) physician and orthopaedic specialists. The results showed that ResNet50 performed better than the ED physician but worse than experienced orthopaedic specialists.

#### 1.2.3. Comparative Study

To demonstrate the uniqueness of this study, we comprehensively conducted a comparative study from seven viewpoints, namely scaphoid segmentation, fracture recognition, surgical treatment recommendation, nonsurgical prognosis status classification, empirical approximation, image type and publication year. Table 1 presents the overall comparison showing that, although there are numerous recent studies on scaphoid segmentation and fracture recognition, none have provided surgical treatment recommendations or nonsurgical prognosis status classification for image types of X-ray, CT or MRI. Furthermore, in this study, a detailed empirical assessment was proposed to approximate the potential solutions for a better diagnosis and treatment of scaphoid fractures. 

The remainder of this paper is structured as follows. The materials and methods for surgical treatment recommendation and prognosis status classification are presented in Section 2. In Section 3, the detailed empirical analysis will be shown. Section 4 presents the research limitations. Lastly, the conclusions and future work are given in Section 5.

## 2. Materials and Methods

### 2.1. Materials

The experimental dataset came from Kaohsiung Chang Gung Memorial Hospital in Taiwan [33], and includes 1210 adult patients with 2438 X-ray images. From this dataset, two subsets were further filtered for surgical treatment recommendation (called the STR set) and nonsurgical prognosis status classification (called the NPS set). In the STR set, 334 images each were set for the surgery and no surgery categories, while 29 images each were set for the avascular necrosis, non-union and union categories in the NPS set. That is, in total, 668 and 87 images were used for surgical treatment recommendation and nonsurgical prognosis status classification, respectively.

### 2.2. Method Framework

The goal of this study was to propose a complementary medicine system for scaphoid fracture recognition and treatment. To reach this goal, two main solutions, namely nonsurgical prognosis status classification and surgical treatment recommendations, are proposed in this paper. Figure 2 and Figure 3 depict the proposed system framework, consisting of offline training and online prediction phases, respectively. 

Offline training: The main aim of this phase is to model the mechanisms by learning from visual images. Therefore, a set of convolutional neural networks are trained with pre-trained models and data augmentations. These networks are the core mechanisms for online prediction.Online prediction: This phase is composed of four modules, namely scaphoid segmentation, fracture recognition, surgical treatment recommendation and nonsurgical prognosis status classification, where the scaphoid segmentation and fracture recognition modules are the fundamental modules supporting the others. The difference among four online modules is just the input data, referring to unknown images, segmented scaphoid images, segmented fracture images and nonsurgical prognosis status images, respectively. Finally, the surgical treatment recommendation will provide the doctor with a surgery reference, while the nonsurgical prognosis status classification will provide the radiologist with a potential prognosis status.

### 2.3. Offline Training

As depicted in Figure 2, the proposed system consists of four offline components, namely data split, data augmentation, transfer learning and convolutional neural networks. In this subsection, the related training components will be presented in detail.

#### 2.3.1. Data Split for Cross Validations

To reveal the model reliability, a cross validation is necessary. In this study, the empirical approximation was set at a 3-fold cross validation. That is, recalling from Figure 2, the data are randomly split into 3 sets (folds) before training. Next, three combinations are generated where each combination consists of two training sets and a testing set. In the experiments, these combinations are examined iteratively. For each combination, a training model is trained and a testing set is then predicted. Finally, three prediction results are derived by three referred training models.

#### 2.3.2. Data Augmentation

Because the gathered data are not enough, the common problem of overfitting might occur to make the results unsatisfactory. Hence, data augmentation is used for dealing with this concern. In this study, for each training fold, an image is augmented by geometric operations of flipping and rotation, where the flipping was set from right to left and the rotation degrees were 15 and −15. Hence, the original data size will be increased by three times compared to the original data size. In the experiments of this study, the augmentation impact was evaluated in detail.

#### 2.3.3. Transfer Learning

In general, a convolutional neural network will incur concerns of efficiency and effectiveness. In terms of efficiency, a general CNN (Convolutional Neural Network) is not efficient because the result is approximated by a number of iterations. That is, the network starts with random convolution filters and is upgraded by iteratively updating the filters. In terms of effectiveness, two further issues can occur in the network. First, there might exist a gap between the training data and unknown data. Second, the random filters are not stable enough to make a reliable prediction, perhaps falling into a local optimal space. To address these concerns, the scenario of using transfer learning in this study consisted of taking a pre-trained model as a support to increase the effectiveness and efficiency. In fact, a pre-trained model named ImageNet [34] has been shown to be effective by many recent studies. However, ImageNet is trained on natural image features, which are different from medical image features. To address this issue, Alzubaidi et al. [35] proposed using medical images as the basis for pre-training the medical image model, which could result in better recognition results compared to using ImageNet. Although this idea is technically sound, this related work did not provide further experimental results and models. Hence, the real effectiveness is not clear. Moreover, because training a large number of medical images requires powerful hardware, the pre-trained model in this study was still ImageNet. To reveal the benefits from using ImageNet, a comprehensive evaluation for this issue was conducted in the experiments. In our implementation, the pre-trained weights provided by ImageNet were used as the initial filters instead of random filters. Next, the network was trained with our experimental data. Finally, the nearly optimal results were approximated more efficiently and effectively. Figure 4 shows how to embed a pre-trained model into a convolutional neural network comprising the components of forward propagation, loss computation, weight updating and backpropagation.

#### 2.3.4. Convolutional Neural Networks

Although the online prediction phase includes the four modules of scaphoid segmentation, fracture recognition, surgical treatment recommendation and nonsurgical prognosis status classification, the core mechanism is the convolutional neural network. In a general convolutional neural network, the image features are filtered by a set of blocks, where each block contains multiple convolutions and pooling. After the blocks, the feature map is flattened into a 1-dimensional vector and is then input into a Multilayer Perceptron (MLP). Finally, the resulting probability for each class is calculated by a softmax function. In this study, we tuned recent CNNs such as ResNet, DenseNet, InceptionNet and EfficientNet, and a comprehensive empirical study was then conducted for surgical treatment recommendation and nonsurgical prognosis status classification. In this section, each involved CNN will be introduced briefly. For a typical CNN, it always suffers from problem of a vanishing gradient as the network depth increases. Therefore, ResNet and DenseNet were proposed to address such problems using identity residuals, while DenseNet is much more complicated than ResNet. As shown in Figure 5a,b, the major difference between ResNet and DenseNet is that DenseNet uses a denser connection, densely connecting each layer to the remaining layers in a forward propagation manner. The main idea of DenseNet is to ensure maximum information flow between layers and to directly connect all layers.

In contrast to ResNet and DenseNet, InceptionNet attempts to make the network wider instead of deeper, as shown in Figure 5c. Based on this attempt, InceptionNet provides multiple convolution threads of different depths to attack the problem of vanishing gradients so as to make the prediction performance more effective and efficient. To consider the concerns of both depth and width, EfficientNet proposes a compound scaling network that approximates a higher accuracy via balancing network depth, width and resolution. The ideas of the above networks motivated us to conduct a comprehensive empirical study, pursuing nearly optimal solutions for surgical treatment recommendation and nonsurgical prognosis status classification. Table 2 depicts the tuned architectures of the compared CNNs and Table 3 depicts the best parameter settings for all the CNNs.

### 2.4. Online Prediction

Referring to Figure 3, the online prediction starts with an unknown X-ray image. Next, fracture recognition is performed to determine whether the scaphoid segmentation there is a fracture or not. If there is a fracture, it will be further recognized by the CNNs for surgical treatment recommendation. If the scaphoid segmentation is from a nonsurgical patient follow-up, the nonsurgical prognosis classifier is used to recognize the current status. Note that, because this study is an extension of a previous work [24], the scaphoid segmentation and fracture recognition are not the major aims of this study. For the details of the scaphoid segmentation and fracture recognition, refer to the reference [24]. Overall, the scaphoid segmentation was implemented using YOLO-v4 [16], reaching an accuracy of around 96%. For fracture recognition, ResNet101 was used as the primary network to recognize the fractures, reaching an accuracy and AUC of around 90.3% and 95%, respectively. Based on these results, the solution proposal will be tested in the succeeding section through a detailed empirical analysis. 

## 3. Results

To achieve the contributions mentioned above, a number of experiments for assessing the effectiveness of recent CNNs were performed in this study. That is, the major intent of the experiments was to elicit the best settings for the transfer learning, data augmentation, batch size and resizing. According to the assessment results, a final recommendation was derived through a comparative study. The terminologies of the compared CNNs are shown in Table 3.

### 3.1. Experimental Data and Measures

The experimental data were gathered from the Kaohsiung Chang Gung Memorial Hospital in Taiwan, and contained two subsets, namely STR and NPS for surgical treatment recommendation and nonsurgical prognosis status classification, respectively. For approximating the best parameter settings, 80% of the data was randomly selected for training and the rest was used for testing. Based on the best approximated settings, for each subset, the original data were randomly divided into three groups for a further 3-fold cross validation. In terms of evaluation measures, accuracy and AUC were the two major metrics used and were based on 4 prediction outcomes, namely True Positive (*TP*), False Positive (*FP*), False Negative (*FN*) and True Negative (*TN*), where *TP*, *TN* and *FP*, *FN* indicate the successful predictions and false predictions, respectively. According to these 4 outcomes, the accuracy can be defined as
(1)$$Accuracy=\frac{TP+TN}{TP+TN+FP+FN}.$$

Also, the AUC indicates the classification performance under different thresholds of true positive rates and false positive rates. Additionally, the other metric used was Improvement representing the performance difference ratio of the proposed method to baseline, which can be defined as
(2)$$Improvement=\frac{RP-RB}{RB},$$
where *RP* and *RB* indicate the results of the proposed method and baseline, respectively.

### 3.2. Evaluations of Transfer Learning without Cross Validations

Because the transfer learning is an important factor for CNN performance, it is necessary to clarify the related impact first. As mentioned above, the major idea of transfer learning is to increase effectiveness and efficiency. Based on the parameter settings in Table 3, Figure 6 and Figure 7 show the comparisons of the CNNs with or without transfer learning on the two main contributions of surgical treatment recommendation and nonsurgical prognosis status classification, respectively. Clearly, the CNNs with transfer learning had better results than those without transfer learning. 

Table 4 summarizes the accuracy improvements that ranged from 13.04% to 45.9%. From these improvements, we can see that on average, the impact of transfer learning for nonsurgical prognosis status classification was larger than that for surgical treatment recommendation. This result might be caused by that fact that three-class recognition is more complex than two-class recognition so the additional pre-trained information is more helpful to three-class recognition. That is, the pre-trained model offers much more valuable information in classifying the prognosis status, especially for the deeper residual-based network ResNet152. These results provide evidence to support using transfer learning for the following evaluations.

### 3.3. Evaluations of Parameter Settings without Cross Validations

Before performing comprehensive assessments of CNNs, the parameters need to be set based on transfer learning. In this evaluation, two main parameters, namely image-size and batch size, were examined to approximate the best settings for the final cross validations.

#### 3.3.1. Parameter Settings for Image Size

As illustrated in Figure 2 and Figure 3, for both training and testing, the scaphoid has to be segmented first. In these scaphoid segmentations, the size ranged from 71 × 94 to 264 × 353. This raises a problem of resizing the input sizes for CNNs. To investigate this problem, the CNNs were evaluated using different input sizes of 112, 140, 168, 196 and 224. Table 5 shows that the best image size was 224 and the best CNNs for the two contributions were InceptionNetV3 and ResNet50, DenseNet121, respectively. Although the best performances of DenseNet201 and InceptionNetV3 were very close for surgical treatment recommendation, the efficiency of InceptionNet was higher than that of DenseNet. In overall, the vertical-based CNNs of ResNet and DenseNet were slightly better than the horizontal-based InceptionNet. Also, the compound scaling network EfficientNet was worse than the other CNNs. Although the overall performances were not very stable, the averaged accuracies using a size of 224 were better than those of the other sizes. Therefore, 224 was the size setting used for further evaluation.

#### 3.3.2. Parameter Settings for Batch Size

In principle, the batch size refers to the number of samples in one training batch, which affects the optimization quality and training speed. Table 6 shows the accuracies of the compared CNNs under different batch sizes for surgical treatment recommendation and nonsurgical prognosis status classification, which show some trends. First, the best CNN for the two contributions were InceV3 and ResNet50, DenseNet121. Second, the best batch size for both contributions was 8. This indicates the better performances will be derived by mini-batch gradient descent learning, moving to the best optimization space smoothly. Third, the deeper residual networks ResNet152, DenseNet201 could not work because of the memory limitation. Fourth, except for a batch size of 8, the CNNs with residuals resulted in better accuracies than those without residuals for both surgical treatment recommendation and nonsurgical prognosis status classification. Clearly, the residuals in the experiments gave important information to effectively avoid the problem of vanishing gradients. According to the results, 8 was set as the batch size for the subsequent evaluations.

### 3.4. Evaluations for Data Augmentation without Cross Validations 

Most of the recent CNNs need a large amount of data to produce a better prediction. However, it is actually not easy to collect a large amount of data especially in the field of biomedicine, causing the problem of overfitting. To address this issue, based on the above approximations of the parameters, the data were enlarged here. After the data augmentation, the data amount for surgical treatment recommendation was expanded from 668 to 2270, while that for nonsurgical prognosis status classification was expanded from 87 to 294. Thereupon, the benefits of the different augmentation paradigms with respect to flipping, $15^{{}^{\circ}}$ rotation and $-15^{{}^{\circ}}$ rotation were investigated by evaluating the compared CNNs. In the following results, the data augmentation will be termed DA for concise representations. 

Figure 8 and Figure 9 show the comparison results of the CNNs with or without data augmentation. Overall, most of the CNNs with data augmentation performed better than those without data augmentation. The observations can be listed as follows. First, the best CNNs for surgical treatment recommendation and nonsurgical prognosis status classification were InceV3 and ResNet50, DenseNet121, respectively. The best settings for both contributions were combining all DA paradigms. Second, as shown in Table 7, the best improvement was 9.46%, which was achieved by ResNet50 for surgical treatment recommendation, while 13.25% was achieved by ResNet50 and DenseNet121 for nonsurgical prognosis status classification. Third, the average improvement for nonsurgical prognosis status classification was slightly smaller than that for surgical treatment recommendation. Overall, the improvement results reveal that more data can bring sufficient information for image recognition.

### 3.5. Comprehensive Evaluations with Cross Validations

After the above basic evaluations, the setting issues were clarified. Accordingly, a comprehensive experiment was conducted to approximate the final recommendations of the solutions for the two contributions. Table 8 shows the accuracies of the compared CNNs with 3-fold cross validations, which depicts several points. First, the best CNNs for the two contributions were DenseNet169 and ResNet50, respectively. Second, the worst classifier was EfficientNet. Third, overall, the residual-based networks performed better than nonresidual-based ones. The possible reason might be that residuals effectively make up for the information loss in the convolutions. In contrast, InceptionNet did not have an improved performance without residuals, and EfficientNet did not succeed in approximating a good balance between the depth and width of a network. Fourth, in comparison with the best results in Table 8, Res50 performed much better than the others for nonsurgical prognosis status classification. For this result, there are two potential explanations. First, as mentioned above, the residuals effectively facilitate the recognition. Second, the recognition does not need a deeper and denser residual network due to the images being not very complicated. On the contrary, for surgical treatment recommendation, the images are so indiscriminative that the recognition needs a deeper and denser residual network such as Den169. In addition to the accuracy, another measure (AUC) was calculated, reaching around 86% and 96% for the two contributions, respectively, exceeding the common baseline of 80% in the field of biomedical information retrieval.

### 3.6. Research Limitations

In above sections, the details of the methodology and evaluations were presented. However, a number of limitations have to be declared here. First, the input image quality was limited by the image capturing devices. Second, the image format was X-ray. Third, the models were generated by the collection. Fourth, the selected bone ages did not include children. For these limitations, discussions will be brought up in the next section of future work.

## 4. Discussion

In order to provide solutions for the two contributions of surgical treatment recommendation and nonsurgical prognosis status classification, a set of evaluations for assessing the effectiveness of recent CNNs were made. However, several remaining issues need to be clarified for a more solid version.

I.In the compared CNNs, two extra components were used as supports to increase the prediction performances, namely transfer learning and data augmentation. This incurred issues for the individual impacts of these components. To address this issue, an ablation study on average improvements using transfer learning and data augmentation for the two contributions was performed, as shown in Table 9. From this table, we can see that transfer learning was more important than data augmentation because the average improvement with transfer learning was more significant than that of data augmentation. The hidden reason could be that the transfer learning supplies confident filter information as the base for the convolutions but the data augmentation does not. This discovery could provide a reference for future researchers.II.In the field of machine learning, an important issue investigated by researchers is the generalization. This can be explained by the testing with or without cross validations in this study. The goal of this cross validation is to clarify whether the performances when using the same settings of transfer learning, data augmentation, epoch, image size and batch size are stable or not while crossing different testing datasets. Table 10 reveals the comparisons between the CNNs with or without cross validations. Overall, the accuracies without cross validations were higher than those with cross validations. For nonsurgical prognosis status classification, ResNet50 had a higher generalization because the related performances were more stable. Finally, the recommended CNN is ResNet50 even though the cross-validation accuracies of ResNet50 and DenseNet121 were equal. This is because ResNet50 is more effective than DenseNet121 in cross validations. In contrast, for surgical treatment recommendation, there exists a bias in the final recommendation. Without cross validations, the final recommendation would be InceV3. However, DenseNet169 is the final recommendation because its cross-validation performance was higher than that of InceV3. This depicts that DenseNet169 is more stable. This is an issue to discuss in this paper.III.In addition to the issues discussed above, an important issue for the effectiveness of embedding an attention mechanism needs to be addressed here. For this issue, a preliminary evaluation was conducted by using a residual attention CNN, namely Attention-56 [36]. In this evaluation, the best classifiers in the above experiments were used as the baselines, including Den169 and Res50. Figure 10 reveals the evaluation results, showing some trends. First, Attention-56 also needs to be pre-trained for better results. Second, the pre-trained Attention-56 could not yield better performances than the selected baselines. The possible interpretations could be that (1) the fracture is not obvious; (2) the appearances are too diverse; or (3) the training data are not enough to make the CNN focus on the areas of interest. In the future, this issue will be investigated for a better result.IV.In summary, the final recommendations for the two contributions are DenseNet169 and ResNet50, respectively, according to the results in Table 8. Furthermore, the performances for nonsurgical prognosis status classification were better than those for surgical treatment recommendation. The potential reason could be that the images for surgical treatment recommendation are indiscriminative, similar to those for nonsurgical treatment recommendation. Therefore, the CNN cannot distinguish them more successfully through convolutional feature filtering.V.In this study, the second contribution was to classify the current prognosis status for nonsurgical images. However, it is more necessary for doctors to know the future status while selecting a treatment. Thus, our next goal is to directly predict the prognosis for unknown images instead of recognizing current statuses. In detail, if the unknown prognosis could be predicted successfully before status occurrences, the doctor can make a more effective treatment decision. For this purpose, a preliminary test using ResNet50 was conducted, which reached as accuracy of around 68%. Although this is not satisfactory, this is just the beginning for this goal. In the future, we will keep improving the performances by using more advanced techniques. Also, this could offer a reference for future research on making a smart medical support system for scaphoid fractures.

## 5. Conclusions

The rapid progress in artificial intelligence has improved the experiences in the field of biomedicine. In particular, over the past few years, biomedical information retrieval using deep learning has played a critical role in effective risk assessments, treatment decisions and prognosis estimations. To these ends, lots of studies have been proposed on effective scaphoid fracture recognition. However, no existing works devoted themselves to surgical treatment recommendations and nonsurgical prognosis status classification. This motivated us to propose a medical support system that includes scaphoid fracture recognition, surgical treatment recommendation and nonsurgical prognosis status classification simultaneously. First, the unknown scaphoid from an X-ray image is segmented. Next, the fracture segmentation is recognized. The related experimental results reached accuracies of 96% and 90% in terms of segmentation and recognition, respectively. Then, based on the scaphoid segmentations, the two contributions of surgical treatment recommendation and nonsurgical prognosis status classification were materialized. For surgical treatment recommendation, the fracture segmentation is further analysed for surgery recommendation, which is helpful to doctors in making an effective decision. For nonsurgical prognosis status classification, the unknown prognosis status scaphoid segmentation is also classified, assisting the doctor in recognizing the current prognosis status. Finally, numerous experiments for evaluating CNNs were performed and an insightful analysis was also provided in this paper. In summary, the recommended CNNs are DenseNet169 and ResNet50, respectively, for the two targeted issues. Even though the best CNNs have been approximated, several issues need to be coped with in the future. First, because the performance was limited in the collected data, federated learning or collaborative learning will need to be utilized for a global performance. Second, the bone age is an issue to investigate further. Third, the attention mechanism will continue to be investigated. Fourth, as mentioned in the discussion section, extensive research on predicting the future statuses of nonsurgical fractures will be conducted. Therefore, a complete medical support system for the diagnosis, treatment and prognosis of scaphoid fractures can be accomplished in the future.

## Figures and Tables

**Figure 1 biomedicines-12-01198-f001:**
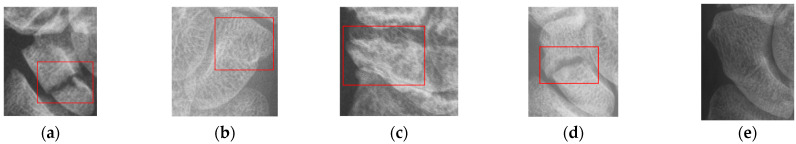
Segmented images of (**a**): surgery; (**b**): no surgery; (**c**): avascular necrosis; (**d**): non-union; (**e**): union statuses for scaphoids where the marked regions are fractures.

**Figure 2 biomedicines-12-01198-f002:**
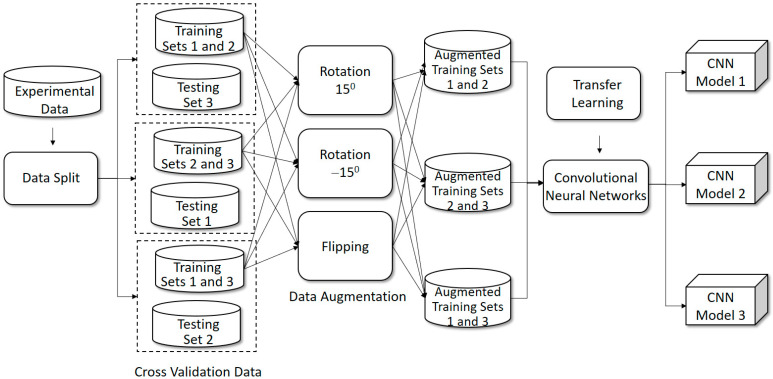
Offline training dataflow.

**Figure 3 biomedicines-12-01198-f003:**
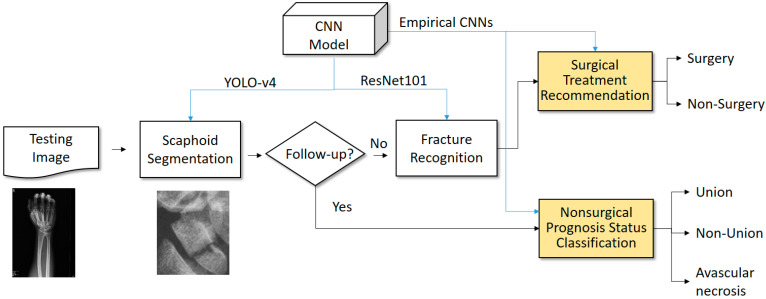
Workflow of online prediction.

**Figure 4 biomedicines-12-01198-f004:**
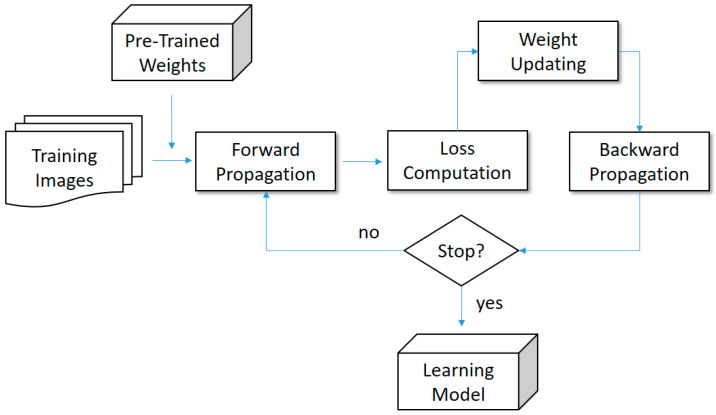
Transfer learning for the proposed convolutional neural networks.

**Figure 5 biomedicines-12-01198-f005:**
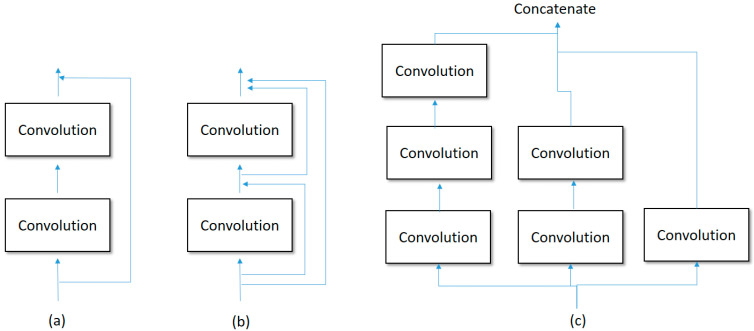
Scenarios for (**a**): ResNet; (**b**): DenseNet; (**c**): InceptionNet.

**Figure 6 biomedicines-12-01198-f006:**
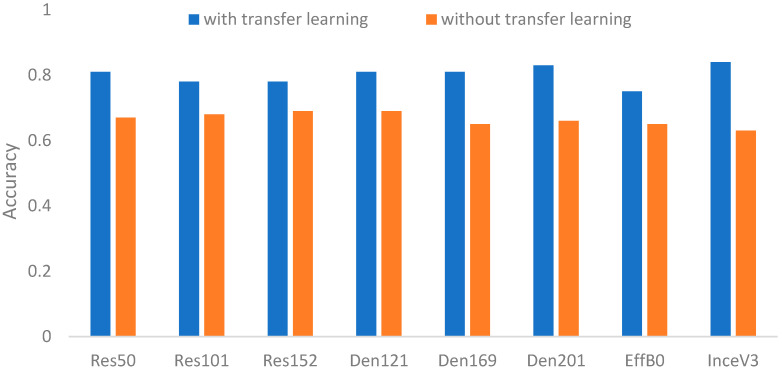
Accuracies of compared CNNs with or without transfer learning for surgical treatment recommendation.

**Figure 7 biomedicines-12-01198-f007:**
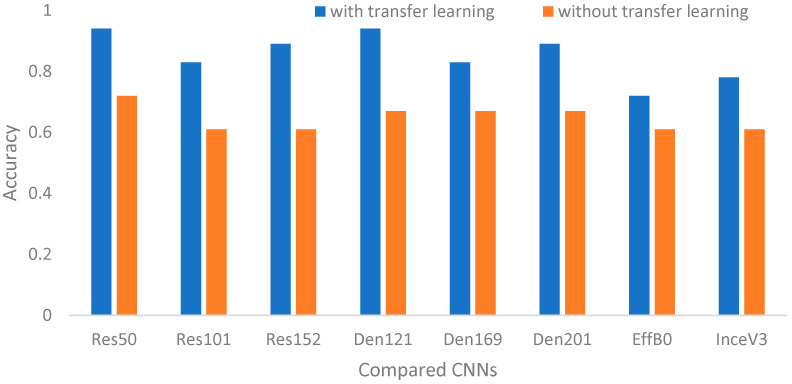
Accuracies of compared CNNs with or without transfer learning for nonsurgical prognosis status classification.

**Figure 8 biomedicines-12-01198-f008:**
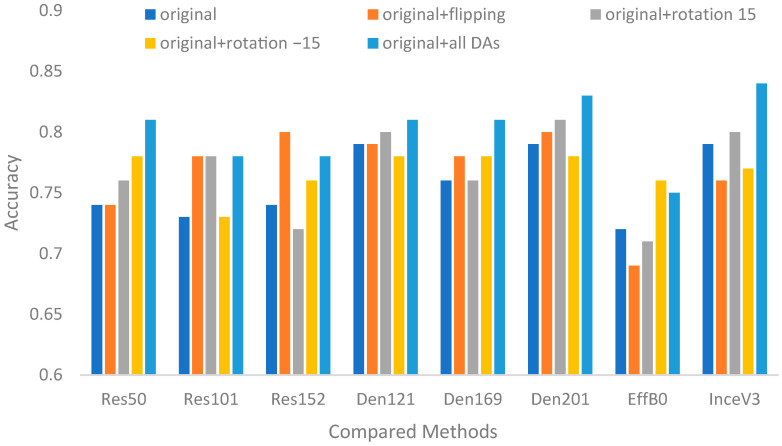
Accuracies of compared CNNs with or without data augmentation for surgical treatment recommendation.

**Figure 9 biomedicines-12-01198-f009:**
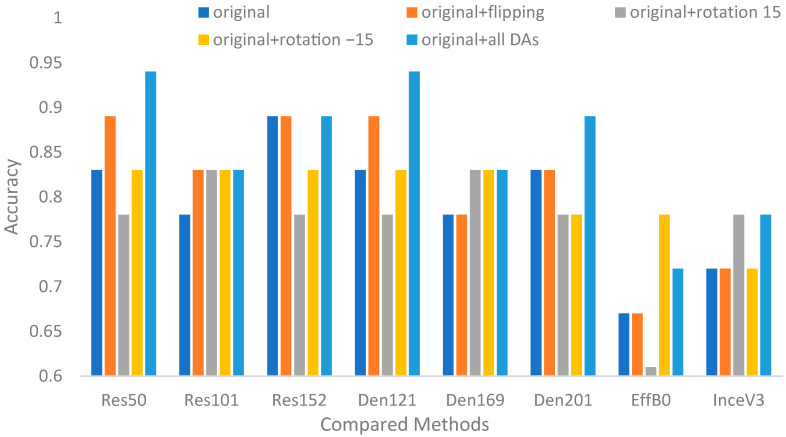
Accuracies of compared CNNs with or without data augmentation for nonsurgical prognosis status classification.

**Figure 10 biomedicines-12-01198-f010:**
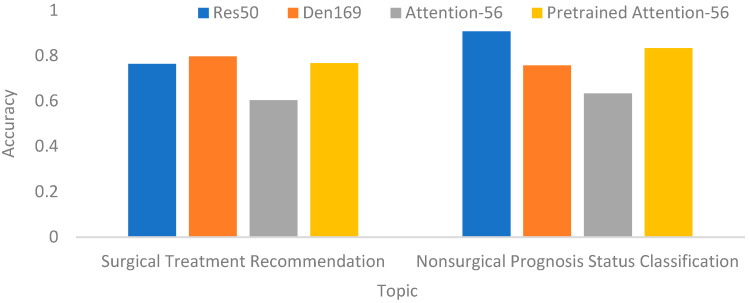
Accuracies of Res50, Den169, Attention-56 and pre-trained Attention-56.

**Table 1 biomedicines-12-01198-t001:** Comparisons of this paper and previous works.

Compared Work	Scaphoid Segmentation	Fracture Recognition	Surgical Treatment Recommendation	Nonsurgical Prognosis Status Classification	Empirical Approximation	Image Type	Publication Year
Tung et al. [24]	Yes, YOLO-v4	Yes, ResNet101 (AUC: 0.95)	No	No	Yes	X-ray	2021
Yoon et al. [25]	Yes, R-CNN	Yes, EfficientNetB3 (AUC: 0.96)	No	No	No	CT&MRI	2021
Li et al. [26]	Yes, YOLO-v3	Yes, MobileNetV3 (AUC: 0.92)	No	No	No	X-ray	2023
Singh et al. [28]	No	Yes, 34-Layer ResNet (AUC: 0.95)	No	No	No	X-ray	2023
Hendrix et al. [29]	Yes, YOLO-v5	Yes, InceptionNetV3 (AUC: 0.88)	No	No	No	X-ray	2023
Yang et al. [30]	Yes, Faster R-CNN	Yes, ResNet152 (AUC: 0.917)	No	No	No	X-ray	2022
Ozkaya et al. [32]	No	Yes, ResNet50 (AUC: 0.84)	No	No	No	CT	2022
Proposed work	Yes, YOLO-v4	Yes, ResNet101 (AUC: 0.95)	Yes, DenseNet 169 (AUC: 0.86)	Yes, ResNet50 (AUC: 0.96)	Yes	X-ray	2024

**Table 2 biomedicines-12-01198-t002:** Architectures of compared CNNs.

	Backbone	#Nodes in Dense Layer 1	Normalization Layer	Dense Layer 2	Flatten	Learning Rate	Optimization Function
Setting	ResNet DenseNet, InceptionNet, EfficientNet	2048	Batch Normalization	Softmax	Global Average Pooling	0.01	SGD (Stochastic Gradient Descent)

**Table 3 biomedicines-12-01198-t003:** Best parameter settings for compared CNNs for two contributions.

CNN	Topic	Batch Size	Resized Size	Epoch
ResNet50 (called Res50)	surgical treatment	8	224	10
	nonsurgical prognosis	8	224	30
ResNet101 (called Res101)	surgical treatment	8	224	10
	nonsurgical prognosis	16	168	30
ResNet152 (called Res152)	surgical treatment	16	168	10
	nonsurgical prognosis	8	224	30
DenseNet121 (called Den121)	surgical treatment	16	112	10
	nonsurgical prognosis	8	224	30
DenseNet169 (called Den169)	surgical treatment	8	224	10
	nonsurgical prognosis	8	224	30
DenseNet201 (called Den201)	surgical treatment	16	168	10
	nonsurgical prognosis	8	224	30
InceptionNetV3 (called InceV3)	surgical treatment	16	112	10
	nonsurgical prognosis	8	168	30
EfficientNetB0 (called EffB0)	surgical treatment	16	196	10
	nonsurgical prognosis	16	112	30

**Table 4 biomedicines-12-01198-t004:** Differences in accuracy between CNNs with or without transfer learning for two main contributions.

Method	Surgical Treatment Recommendation	Nonsurgical Prognosis Status Classification
Res50	0.208955	0.305556
Res101	0.147059	0.360656
Res152	0.130435	0.459016
Den121	0.173913	0.402985
Den169	0.246154	0.238806
Den201	0.257576	0.328358
EffB0	0.153846	0.180328
InceV3	0.333333	0.278689

**Table 5 biomedicines-12-01198-t005:** Accuracies of compared CNNs under different image sizes for surgical treatment recommendation and nonsurgical prognosis status classification.

	Surgical Treatment Recommendation					Nonsurgical Prognosis Status Classification				
CNN\Size	112	140	168	196	224	112	140	168	196	224
Res50	0.76	0.74	0.76	0.76	0.81	0.83	0.83	0.83	0.83	0.94
Res101	0.74	0.74	0.77	0.74	0.78	0.78	0.83	0.89	0.78	0.83
Res152	0.8	0.8	0.81	0.78	0.78	0.83	0.83	0.78	0.83	0.89
Den121	0.81	0.78	0.78	0.78	0.81	0.78	0.78	0.72	0.78	0.94
Den169	0.77	0.8	0.77	0.77	0.81	0.78	0.72	0.83	0.78	0.83
Den201	0.76	0.77	0.81	0.77	0.83	0.83	0.83	0.83	0.83	0.89
EffB0	0.73	0.77	0.74	0.79	0.75	0.83	0.78	0.67	0.67	0.72
InceV3	0.78	0.74	0.75	0.75	0.84	0.67	0.83	0.83	0.72	0.78

**Table 6 biomedicines-12-01198-t006:** Accuracies of compared CNNs under different batch sizes for surgical treatment recommendation and nonsurgical prognosis status classification.

	Surgical Treatment Recommendation				Nonsurgical Prognosis Status Classification			
CNN\Size	8	16	24	32	8	16	24	32
Res50	0.81	0.74	0.75	0.79	0.94	0.83	0.78	0.78
Res101	0.78	0.76	0.76	0.78	0.83	0.83	0.72	0.72
Res152	0.78	0.78	0.76	Out of Memory	0.89	0.83	0.78	Out of Memory
Den121	0.81	0.8	0.79	0.78	0.94	0.78	0.72	0.72
Den169	0.81	0.8	0.77	0.78	0.83	0.78	0.78	0.72
Den201	0.83	0.8	0.77	Out of Memory	0.89	0.78	0.72	Out of Memory
EffB0	0.75	0.78	0.72	0.75	0.72	0.78	0.72	0.72
InceV3	0.84	0.78	0.72	0.78	0.78	0.72	0.72	0.67

**Table 7 biomedicines-12-01198-t007:** Differences in accuracy between CNNs with or without data augmentation for two main contributions.

Method	Surgical Treatment Recommendation	Nonsurgical Prognosis Status Classification
Res50	0.094595	0.13253
Res101	0.068493	0.064103
Res152	0.054054	0
Den121	0.025316	0.13253
Den169	0.065789	0.064103
Den201	0.050633	0.072289
EffB0	0.041667	0.074627
InceV3	0.063291	0.083333

**Table 8 biomedicines-12-01198-t008:** Accuracies of compared CNNs with 3-fold cross validations.

Method	Surgical Treatment Recommendation	Nonsurgical Prognosis Status Classification
Res50	0.763333	0.906667 *
Res101	0.766667	0.776667
Res152	0.75	0.816667
Den121	0.766667	0.776667
Den169	0.79667 *	0.75667
Den201	0.773333	0.76
EffB0	0.726667	0.666667
InceV3	0.77	0.796667

Note that * denotes the best accuracy.

**Table 9 biomedicines-12-01198-t009:** Comparisons of average improvements using transfer learning and data augmentation for two contributions.

Setting	Method	Transfer Learning	Data Augmentation
Surgical Treatment Recommendation	ResNet	0.16215	0.072381
	DenseNet	0.225881	0.047246
	EfficientNet	0.153846	0.041667
	InceptionNet	0.333333	0.063291
Nonsurgical Prognosis Status Classification	ResNet	0.375076	0.065544
	DenseNet	0.323383	0.089641
	EfficientNet	0.180328	0.074627
	InceptionNet	0.278689	0.083333

**Table 10 biomedicines-12-01198-t010:** Generalization study results on accuracies of testing with or without cross validations for two contributions.

Setting	Method	with Cross Validations	without Cross Validations
Surgical Treatment Recommendation	Res50	0.76	0.81
	Res101	0.77	0.78
	Res152	0.75	0.78
	Den121	0.77	0.81
	Den169	0.8	0.81
	Den201	0.77	0.83
	EffB0	0.73	0.75
	InceV3	0.77	0.84
Nonsurgical Prognosis Status Classification	Res50	0.91	0.94
	Res101	0.76	0.83
	Res152	0.82	0.89
	Den121	0.72	0.94
	Den201	0.76	0.83
	Den169	0.74	0.89
	EffB0	0.65	0.72
	InceV3	0.8	0.78

## Data Availability

The patients of this study did not give written consent for their data to be shared publicly, so due to the sensitive nature of the research, the supporting data are not available.

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
