# Peer review of "Deep Learning-Based Surgical Treatment Recommendation and Nonsurgical Prognosis Status Classification for Scaphoid Fractures by Automated X-ray Image Recognition"

_biomedicines, 2024, doi:10.3390/biomedicines12061198_

Round 1
Reviewer 1 Report
Comments and Suggestions for Authors
The manuscript “Deep Surgical Treatment Recommendation and Nonsurgical Prognosis Status Classification for Scaphoid Fractures by Automated X-ray Image Recognition“ by Su et al. reports solutions by a comprehensive empirical study for assessing the effectiveness of recent deep learning on surgical treatment recommendation and nonsurgical prognosis image classification. This work is well designed and could be accepted after revision. Here are the comments and suggestions:
1. English of this manuscript should be improved.
2. Some more important results can be added to the Abstract.
3. Abbreviations should be defined before their first use.
4. Some Tables can be combined or moved to the supporting information to make this work more concise.
Comments on the Quality of English Language1. English of this manuscript should be improved.
2. Abbreviations should be defined before their first use.
Author Response
The authors are grateful for the reviewers’ helpful comments that are valuable in improving this paper. We have revised the manuscript as follows.
Revision made in accordance with comments by Reviewer No.1
The manuscript “Deep Surgical Treatment Recommendation and Nonsurgical Prognosis Status Classification for Scaphoid Fractures by Automated X-ray Image Recognition “ by Su et al. reports solutions by a comprehensive empirical study for assessing the effectiveness of recent deep learning on surgical treatment recommendation and nonsurgical prognosis image classification. This work is well designed and could be accepted after revision. Here are the comments and suggestions:
- English of this manuscript should be improved.
Answer: Thanks for this comment. We have made a detailed check for grammar, typo and presentation.
- Some more important results can be added to the Abstract
Answer: Thanks for this comment. According to this comment, we added the performance improvements using transfer learning and data augmentation into the abstract (P. 1). Moreover, we concluded the final recommendation methods. Please refer to the following for a quick review.
“Further, the methods with transfer learning and data augmentation can bring out obvious improvements, on average, reaching {21.9%, 28.9%} and {5.6%, 7.8%} for {surgical treatment recommendation, nonsurgical prognosis image classification}, respectively. Based on the experimental results, the recommended methods in this paper are DenseNet169 and ResNet50 for surgical treatment recommendation and nonsurgical prognosis image classification, respectively.”
- Abbreviations should be defined before their first use.
Answer: Thanks for this comment. For this comment, we checked the paper and defined the abbreviations carefully. In the following, we list the defined terms for a quick review.
- AUC (Area Under Curve) in abstract;
- AUC (Area Under Curve) in P.3;
- Computer Vision (CV) in P.3;
- YOLO (You Only Look Once) in P.3;
- Non-Maximum Suppression (NMS) in P. 3;
- Feature Pyramid Network (FPN) in P. 3;
- CT (Computed Tomography) in P. 4;
- MRI (Magnetic Resonance Imaging) in P. 3;
- R-CNN (Region-based Convolutional Neural Networks) in P. 3;
- ED (Emergency Department) in P. 4;
- Surgical Treatment Recommendation (called STR set) in P. 5;
- Nonsurgical Prognosis Status classification (called NPS set) in P. 5;
- CNN (Convolutional Neural Networks) in P. 6;
- Multilayer Perceptron (MLP) in P. 7;
- SGD (Stochastic Gradient Descent) in Table 2, P.8;
- True Positive (TP), False Positive (FP), False Negative (FN) and True Negative (TN) in P. 9;
- “RP and RB indicate the results of proposed method and baseline” in P. 9;
- “data augmentation will be termed as DA” in P. 12;
- Some Tables can be combined or moved to the supporting information to make this work more concise.
Answer: Thanks for this comment. Based on this comment, we made modifications as follows.
- First, we moved the terminologies from Table 4 to Table 3. Therefore, the original Table 4 was deleted.
- Second, we combined the original Tables 6 and 7 into the new Table 5.
- Third, we combined the original Tables 8 and 9 into the new Table 6.

Reviewer 2 Report
Comments and Suggestions for Authors
Thank you for sharing the research study. However, the value and novelity are very limited. Here are my comments below.
1. The impact of scaphoid segmentation and fracture recognition are not strong. Please rewrite the introduction.
2. Areas of fracture should be highlighted in figure 1.
3. Year of publication, machine learning approach and performance should be included in Table 1.
4. Figure 2 has issue. CNN block indicates predicton to scaphoid segmentation and fracture recongnition. This figure is a mixture of data flow and system block diagrm. It should be separated.
5. The dataset of each training and test data should be shown in new figure of dataflow.
6. Althrough Imagnet can commonly used for pre-train purpose, Grey-Mednet is more specific for medical imaging pre-training weight. Please make comment.
7. Figure 4 arrow direction are inconsistent.
8. Attention mechanism is widely used in deep learning. Residual attention convolutional neural network such as attention-56 should be involved in this study.
9. The role of YoloV4 is not clear. It is not shown in Figure 2. It seems that all images needs to segment out before classification. A simple image should be provided.
10. Evaluation strategy such as with and without validation has not summarized in method section.
11. It is interesting to see that ResNet-50 outperforms the rest of net in general. However, it has litttle explaination on this.
12. The research limitation should not be on the hardware issue. The dataset issues must contribute much more. The research limitation must be written.
Author Response
The authors are grateful for the reviewers’ helpful comments that are valuable in improving this paper. We have revised the manuscript as follows.
Revision made in accordance with comments by Reviewer No.2
- The impact of scaphoid segmentation and fracture recognition are not strong. Please rewrite the introduction.
Answer: Thanks for this comment. For this comment, we added an interpretation into the introduction (P. 2). Please kindly find below for a quick review.
“Although this paper aims at the surgical treatment recommendation and nonsurgical prognosis status classification, the scaphoid segmentation and fracture recognition are very important in this paper. Without the successful scaphoid segmentation and fracture recognition, it is not easy to achieve high qualities of surgical treatment recommendation and nonsurgical prognosis status classification.”
- Areas of fracture should be highlighted in figure 1
Answer: Thanks for this comment. We highlighted the fracture to address this issue. Please refer to Figure 1 in Section 1 (P. 2). Also, please kindly find below for a quick review.
- Year of publication, machine learning approach and performance should be included in Table 1.
Answer: Thanks for this comment. We have added information of publication year, machine learning approach and performance into Table 1 (Subsection 1.2.3, P. 4). Please kindly find below for a quick review.
- Figure 2 has issue. CNN block indicates prediction to scaphoid segmentation and fracture recognition. This figure is a mixture of data flow and system block diagram. It should be separated.
Answer: Thanks for this comment. Based on this valuable comment, we split the system flow into two subflows, namely training dataflow and workflow of prediction, which can be seen in Figures 2 and 3 (Subsection 2.2, P. 5). Please kindly find below for a quick review.
- The dataset of each training and test data should be shown in new figure of dataflow.
Answer: Thanks for this comment. As concern of above Question 4, we divided the flow into two granular flows to make the framework clearer, namely training dataflow and workflow of prediction, which can be seen in Figures 2 and 3 (Subsection 2.2, P. 5). In Figure 2, the data are randomly split into 3 sets (folds) and 3 combinations where each combination contains two training sets and a testing set. Please kindly find above figures for a quick review.
- Although Imagnet can commonly used for pre-train purpose, Grey-Mednet is more specific for medical imaging pre-training weight. Please make comment.
Answer: Thanks for this comment. Yes, we agreed with this point and conducted an additional paragraph for clarifying this issue (Subsection 2.3.3, P. 6). Also, this related work was cited as a new reference [35]. Please kindly find above figures for a quick review.
“Actually, a pre-trained model named ImageNet [34] has been approved to be effective by many recent researches. However, ImageNet is trained on natural images, which are different from medical image features. To aim at this issue, Alzubaidi et al. [35] proposed using medical images as the basis for pre-training the medical image model, which could result in better recognition results than using ImageNet. Although this idea is technically sound, this related work did not provide further experimental results and models. Hence the real effectiveness is not clear. Besides, because training a large number of medical images requires powerful hardware, the pre-trained model in this paper is still ImageNet. To reveal the benefit from ImageNet, a comprehensive evaluation for this issue will be conducted in the experiments.”
- Figure 4 arrow direction are inconsistent.
Answer: Thanks for this comment. This is our careless mistake. We have corrected this mistake in Figure 5 (Subsection 2.3.4, P. 7). Please kindly find below for a quick review.
- Attention mechanism is widely used in deep learning. Residual attention convolutional neural network such as attention-56 should be involved in this study.
Answer: Thanks for this comment. For this concern, we added a new discussion with an additional evaluation (Subsection 3.6, P. 15). The results show that, the attention-56 cannot bring out better results than the compared baselines. However, we will keep investigating this issue for a better result. Also, the related work was cited as a new reference [36] and the lasted investigation intent was added into our future work. Please kindly find below for a quick review.
“In addition to above discussed issues, a high-value issue for effectiveness of embedding attention mechanism needs to be addressed here. For this issue, a preliminary evaluation was conducted by using a residual attention CNN, namely Attention-56 [36]. In this evaluation, the best classifiers in above experiments were used as the baselines, including Den169 and Res50. Figure 10 reveals the evaluation results, delivering some aspects. First, the Attention-56 also needs to be pre-trained for better results. Second, the pre-trained Attention-56 cannot still bring better performances than the selected baselines. The possible interpretations are: 1) the fracture is not obvious; 2) the appearances are too diverse; 3) the training data are not enough to focus on the attentions. In the future, this issue will be investigated for a better result.”
- The role of YoloV4 is not clear. It is not shown in Figure 2. It seems that all images needs to segment out before classification. A simple image should be provided.
Answer: Thanks for this comment. For this comment, we modified the prediction flow, and added illustrative images and CNN tags into Figure 3 (Subsection 2.2, P. 5). Please kindly find below for a quick review.
- Evaluation strategy such as with and without validation has not summarized in method section.
Answer: Thanks for this comment. According to this comment, we added a new Subsection 2.3.1 entitled as “Data Split for Cross Validation” (P. 6). Please kindly find below for a quick review.
“2.3.1. Data Split for Cross Validation
To reveal the model reliability, cross validation is necessary. In this paper, the empirical approximation is set by a 3-fold cross validation. That is, as shown in Figure 2, the data are randomly split into 3 sets (folds) before training. Next, three combinations are generated where each combination consists of two training sets and a testing set. In training, each fold in a combination serves as a testing set, while the others are used for training. After testing, three prediction results are derived by three referred training models.”
- It is interesting to see that ResNet-50 outperforms the rest of net in general. However, it has little explanation on this.
Answer: Thanks for this comment. Actually, the similar performances are shown in our other cancer recognition such as Melanoma. To clarify this issue, we added an interpretation into Subsection 3.5 (P. 13). Please kindly find below for a quick review.
“Table 8 shows the accuracies of compared CNNs with 3-fold cross validations, which depicting several points. First, the best CNNs for two contributions are DenseNet169 and ResNet50, respectively. Second, the worst classifier is EfficientNet. Third, in overall, the residual-based networks perform better than nonresidual-based ones. The possible reason might be that residuals effectively make up the information loss in convolutions. In contrast, the InceptionNet does not work superior without residuals, and the EfficientNet does not succeed to approximate a good balance of depth and width of a network. Fourth, in comparison with the best results in Table 8, Res50 is much better than the others for nonsurgical prognosis status classification. For this result, there are two potential explanations. First, as mentioned in above, the residuals effectively facilitate the recognition. Second, the recognition does not need a deeper and denser residual network due to the images are not so complicated. On the contrary, for surgical treatment recommendation, the images are indiscriminative that the recognition needs a deeper and denser residual network such as Den169.”
- The research limitation should not be on the hardware issue. The dataset issues must contribute much more. The research limitation must be written.
Answer: Thanks for this comment. Based on this comment, we dropped the hardware issue and rewrote the research limitation (Section 4, P. 15). Please kindly find below for a quick review.
“In above sections, details of methodology and evaluations have been presented. However, a number of limitations have to be declared here. First, the input image quality is limited in the same image capturing devices. Second, the image format is X-ray. Third, the models are generated by the collection. Fourth, the aimed bone age did not include children.”

Round 2
Reviewer 1 Report
Comments and Suggestions for Authors
It seems more acceptable now.
Author Response
The authors are grateful for the reviewers’ helpful comments that are valuable in improving this paper. We have revised the manuscript as follows.
Revision made in accordance with comments by Reviewer No.1
- It seems more acceptable now.
Answer: Thank you very much for the positive comment.

Reviewer 2 Report
Comments and Suggestions for Authors
Thanks for addressing all questions. The figure quality and English should be reviewed.
Comments on the Quality of English LanguageSuggest using English editing service
Author Response
The authors are grateful for the reviewers’ helpful comments that are valuable in improving this paper. We have revised the manuscript as follows.
Revision made in accordance with comments by Reviewer No.2
- Thanks for addressing all questions. The figure quality and English should be reviewed.
Answer: Thank you very much for this comment. For the figure quality, we updated all figures with high-quality ones. Also, we attached the supplementary file with all editable sources. For writing, we carefully checked the typo, format, grammar and linguistic presentation several times. All errors were corrected and the unclear presentations were rewritten. Please kindly refer to the modifications in red.
